# Glucose Transporters as a Target for Anticancer Therapy

**DOI:** 10.3390/cancers13164184

**Published:** 2021-08-20

**Authors:** Monika Pliszka, Leszek Szablewski

**Affiliations:** Chair and Department of General Biology and Parasitology, Medical University of Warsaw, 5 Chalubinskiego Str., 02-004 Warsaw, Poland; monika.pliszka@wum.edu.pl

**Keywords:** GLUT proteins, sodium-dependent glucose cotransporters, cancers therapy

## Abstract

**Simple Summary:**

For mammalian cells, glucose is a major source of energy. In the presence of oxygen, a complete breakdown of glucose generates 36 molecules of ATP from one molecule of glucose. Hypoxia is a hallmark of cancer; therefore, cancer cells prefer the process of glycolysis, which generates only two molecules of ATP from one molecule of glucose, and cancer cells need more molecules of glucose in comparison with normal cells. Increased uptake of glucose by cancer cells is due to increased expression of glucose transporters. However, overexpression of glucose transporters, promoting the process of carcinogenesis, and increasing aggressiveness and invasiveness of tumors, may have also a beneficial effect. For example, upregulation of glucose transporters is used in diagnostic techniques such as FDG-PET. Therapeutic inhibition of glucose transporters may be a method of treatment of cancer patients. On the other hand, upregulation of glucose transporters, which are used in radioiodine therapy, can help patients with cancers.

**Abstract:**

Tumor growth causes cancer cells to become hypoxic. A hypoxic condition is a hallmark of cancer. Metabolism of cancer cells differs from metabolism of normal cells. Cancer cells prefer the process of glycolysis as a source of ATP. Process of glycolysis generates only two molecules of ATP per one molecule of glucose, whereas the complete oxidative breakdown of one molecule of glucose yields 36 molecules of ATP. Therefore, cancer cells need more molecules of glucose in comparison with normal cells. Increased uptake of glucose by these cells is due to overexpression of glucose transporters, especially GLUT1 and GLUT3, that are hypoxia responsive, as well as other glucose transport proteins. Increased expression of these carrier proteins may be used in anticancer therapy. This phenomenon is used in diagnostic techniques such as FDG-PET. It is also suggested, and there are observations, that therapeutic inhibition of glucose transporters may be a method in treatment of cancer patients. On the other hand, there are described cases, in which upregulation of glucose transporters, as, for example, NIS, which is used in radioiodine therapy, can help patients with cancer. The aim of this review is the presentation of possibilities, and how glucose transporters can be used in anticancer therapy.

## 1. Introduction

Cancer is a worldwide health problem. Several cancers, such as pancreatic, hepatic, colorectal, prostate, breast, lung cancers, and osteosarcoma in pediatric patients, are common. On the other hand, there are also rare and very rare cancers, such as acinar cell carcinoma, among pancreatic cancers, pancreatoblastoma, and penile cancers. Many of these cancers are highly lethal, such as hepatocellular carcinoma, pancreatic cancers, epithelial ovarian, and breast cancers among women, lung cancer, and so on. Hepatocellular carcinoma, ovarian cancer, lung cancer, undifferentiated thyroid carcinomas, oral squamous cell carcinoma, or esophageal cancer have worst prognosis, and patients with these cancers have a significant poorer overall survival (OS), decreased disease-free survival, and decreased metastasis-free survival. There are more aggressive tumors, such as primary brain tumors (for example, glioblastoma) and less aggressive, as, for example, some types of thymic carcinoma. There are the highly malignant tumors, for example melanoma, whereas urothelial papilloma has low malignant potential, as well as only a small fraction of thyroid cancers being malignant. Osteosarcoma is highly metastatic cancer, whereas ameloblastoma, originating from odontogenic tissues, has no tendency to metastasize. Several factors influence these properties. One of these is the expression of glucose transporters in cancer cells. Obtained results showed that, in many cases, characteristics of cancers were correlated with levels of glucose transporters expression. It was found, for example, that overexpression of GLUT1 and GLUT3 increases aggressiveness and invasiveness of tumors; its malignance, the tendency to metastasize, decreases metastasis-free survival, causes a significantly poorer overall survival, and predicts worse prognosis in patients [1].

What is a cause of such a high correlation between the expression of glucose transporters and characteristics of cancers? Mammalian cells prefer glucose as a source of energy. In the presence of oxygen, processes of glycolysis, oxidative decarboxylation, Krebs cycle, and oxidative phosphorylation cause a complete breakdown of glucose. In this way, human cells yield 36 molecules of ATP from one molecule of glucose. The growth of cancer causes cancer cells to become hypoxic, and hypoxia is a hallmark of cancer. It was found that cancer cells exhibit a specific type of metabolism. These cells secrete lactate, which is an end-product of glycolysis [2]. Glycolysis generates only two molecules of ATP from one molecule of glucose. In the presence of oxygen, cancer cells prefer the process of glycolysis. This phenomenon is called the “Warburg effect” of “aerobic glycolysis” [2,3]. Glycolysis regulates genes that are involved in the hypoxia-induced metabolic switch [4]. Because glycolysis yields less molecules of ATP than oxidative breakdown of glucose, cancer cells need more molecules of glucose in comparison with normal cells. Increased uptake of glucose by cancer cells is due to increased expression of glucose transporters in these cells. This process is observed especially in the case of GLUT1 and GLUT3. These GLUT proteins are hypoxia responsive [5]. Upregulation of glucose transporters is used in diagnostic techniques such as FDG-PET. It is also suggested that therapeutic inhibition of glucose transporters by natural or synthetic inhibitors, by antibodies anti-glucose transporters, siRNA, shRNA, and so on, may be a method in the treatment of cancer patients. On the other hand, there are described cases, in which upregulation of a glucose transporter, such as, for example, NIS, which is used in radioiodine therapy, can help patients with cancers.

The aim of this review is the presentation of possibilities of how glucose transporters can be used in anticancer therapy.

## 2. Human Glucose Transporters

Glucose is hydrophilic in nature. Therefore, it cannot penetrate the lipid bilayer of plasmalemma. Transport of glucose into cells across the cellular membranes requires specific membrane carrier proteins, so-called glucose transporters. Glucose transporters belong to the major facilitator superfamily (MFS), which contains 74 families of membrane transporters. To date, more than 10,000 members have been sequenced.

There are three distinct families of genes coding glucose transporters in humans: SLC2A genes that code sodium-independent glucose transporters (facilitated transport, GLUT proteins), SLC5A genes, that code sodium-dependent glucose symporters (secondary active transport, SGLT proteins), and SLC50A genes, which code SWEET protein [6]. However, these carrier proteins are called “glucose transporters”, and they transport several different molecules, such as fructose, mannose, galactose, vitamins, myo-inositol, urate, glucosamine, iodide, and ions such as SCN^−^, NO_3_^−^, Br^−^, and so on, but not only glucose. They may also play a role as glucose sensors, as a receptor for human T cell leukemia virus type-1 (HTLV), and as an autoimmune modifier gene [6,7]. Most cells express different glucose transporters in the dependence on specific metabolic requirements.

### 2.1. Sodium-Independent Glucose Transporters

In humans, there are 14 members of the sodium-independent glucose transporters (Table 1). There are GLUT1–GLUT14, encoded by SLC2A1–SLC2A14 genes, respectively [8]. These glucose transporters contain 12 hydrophobic α-helical transmembrane (TM) domains that are connected by the hydrophilic loop between TM6 and TM7 of the GLUT [9]. GLUT proteins contain a short intracellular N-terminal segment and a large C-terminal segment. GLUTs also contain a single site for glycosylation on the exofacial end, either in the large loop between TM1 and TM2 or between TM9 and TM10 [10]. Comparison of sequences of all GLUTs shows better conserved sequences in the putative transmembrane regions, whereas the sequences in the loops and the C- and N-termini are most divergent [11]. Based on the phylogenetic analysis of sequence similarity, GLUT proteins are divided into three classes [11]. Class I contains GLUT1–GLUT4 and GLUT14, Class II comprises GLUT5, GLUT7, GLUT9, and GLUT 11, Class II includes GLUT6, GLUT8, GLUT10, GLUT12, and GLUT13 (HMIT) [12]. All GLUT proteins are facilitative transporters, except for GLUT13, which is an H^+^/myo-inositol symporter [13].

### 2.2. Sodium-Dependent Glucose Symporters

The second family of glucose transporters are sodium-dependent glucose symporters (Table 2). They are the SGLT or sodium/substrate symporters family (SSSF), which contains over 450 members. These membrane proteins are coded by SLC5A genes [6,14,15]. In humans, there are described 12 genes encoded these symporters: SGLT1–SGLT6, SMIT1, NIS, SMVT, CHT1, SMCT1, and SMCT2. These symporters contain 580–718 amino acids, with a predicted mass of 60–80 kDa. Ten of the sodium/symporters contain 14 transmembrane α-helices, whereas NIS and SMCT1 both lack TM14 [16]. Both the hydrophilic N- and C-termini are located on the extracellular side of the cell membrane [6]. SGLTs are highly glycosylated proteins; however, glycosylation is not required in the functioning of these symporters. Eleven of them are sodium/glucose cotransporters, whereas SGLT3, encoded by SLC5A4 gene, is a glucose sensor [17]. SGLTs transport across plasma membrane substrates such as monosaccharides, disaccharides, myo-inositol, pyruvate, nicotinate, vitamins, and ions such as I^−^, ClO_4_^−^, SCN^−^, NO_3_^−^, and Br^−^. SGLT6 also interacts with the immune-related genes and plays a role as an autoimmune gene. CHT1 is a Cl^-^-dependent Na^+^/choline cotransporter [18].

### 2.3. SWEET Protein

SWEET is a new class of glucose transporters. Human SWEET1 is encoded by SLC50A1 gene. It comprises 221 amino acids with a molecular weight of 25 kDa. SWEET1 contains seven predicted transmembrane domains with two internal triple-helix-bundles. They are connected by an inversion linker helix (TM4), creating the translocation pathway (3 + 1 + 3 configuration). It transports mono- and disaccharides across vacuolar and plasma membranes. Human SWEET1 is predominantly expressed in the Golgi complex. It is suggested that it serves to supply glucose for lactose synthesis in the mammary gland [19].

## 3. Glucose Transporters as a Therapeutic Target in Cancer

In several cancers, upregulation of glucose transporters was observed as compared to healthy tissues and cells. For example, an increased level of GLUT1 expression was observed in hepatocellular carcinoma [20], pancreatic tumors [21], prostate cancer [22], cervical squamous cell carcinoma [23], and several others. GLUT2 is overexpressed in the hepatocellular carcinoma cells [24] and colorectal cancer [25]. Upregulation of GLUT3 was observed in the case of papillary thyroid carcinoma [26] and oral squamous cell carcinoma [27]. There are results that revealed the overexpression of remaining glucose transporters, such as GLUT4, GLUT5, GLUT6, and GLUT12. In cancer cells, changes in SGLTs expression were also observed, such as in the case of SGLT1 and NIS [1]. There are also observations that expression of glucose transporters is decreased in cancers. For example, GLUT2 is downregulated in renal cell carcinoma and chromophobes’ renal cell carcinoma [28], downregulation of GLUT4 was detected in clear cell renal cell carcinoma [29] and in pancreatic cancers [30], and the level of GLUT9 expression is decreased in prostate cancers [31].

### 3.1. Role of Glucose Transporters in the Diagnostic Technique

As mentioned earlier, cancer cells require many more molecules of glucose than normal cells. Increased uptake of glucose by cancer cells is the basis of the diagnostic technique, known as ^18^F-deoxy-glucose positron emission tomography (FDG-PET), which is the noninvasive diagnostic and prognostic technique. In this technique, a tracer is used ^18^F-2-fluoro-2-deoxy-D-glucose (^18^F-FDG), which contains a nonmetabolized analog of glucose, 2-deoxy-D-glucose. After an intravenous administration of ^18^F-FDG, the tracer enters the cells through GLUT proteins, preferentially in tumors, where it is accumulated within cells. Higher uptake of ^18^F-FDG is correlated with more aggressive and advanced-stage tumors [32]. This diagnostic technique is a widely used method, especially in the field of oncology.

There were investigated correlations between the FDG uptake and the expression of glucose transporters GLUT1–GLUT5. Obtained results showed that significantly elevated expression levels of GLUT1 and GLUT3 are a factor contributing to the accumulation of FDG in malignant tumors [33]. The correlation between the expression of GLUT1 and uptake of ^18^F-FDG was observed in the pulmonary metastatic tumors; however, the level of GLUT1 was higher in the pulmonary metastatic tumors in comparison with primary lung cancer [34]. A strong positive correlation between the expression of GLUT1 and uptake of labeled FDG was observed in lung cancer [34] and in the primary non-small cell lung cancer [35]. Investigations of 22 different cancers revealed the highest correlation between the level of GLUT1 expression and FDG uptake in pancreatic cancer, and the lowest in the case of colorectal cancer [36]. The association between the level of GLUT1 and GLUT3 expression was detected in hepatocellular carcinoma [37], thyroid cancer [38], and in several other cancers [39].

On the other hand, a lack of positive correlation was found between the expression of GLUT1 and uptake of ^18^F-FDG in the case of human melanoma cells [40]. There is also a lack of correlation between the labeled tracer uptake and the level of GLUT1 and GLUT3 expression in human breast cell lines [41]. Similarly, in schwannomas, microvascular density and vascular permeability are correlated more with the uptake of labeled FDG than with the expression of GLUT1 and GLUT3.

This diagnostic and prognostic technique is used in several cancers, such as lung cancer [42], esophageal carcinoma [43], and head and neck squamous cell carcinomas [44]. This diagnostic technique is used not only in oncology, but also in the granulomatous disease, autoimmune disease, and in the sites of infection and in inflammation [45].

### 3.2. GLUT Proteins as a Target in Anticancer Therapy

As described earlier, cancer cells cause changes in the expression of genes that code HIF-1, hexokinase 2 (HK2), phosphoglucose isomerase (PG), glyceraldehyde 3-phosphate dehydrogenase, and glucose transporters that are involved in the hypoxic conditions within tumors. Therefore, it is suggested that hypoxia-regulated genes may be a therapeutic target in anticancer therapy [46]. GLUT proteins, such as GLUT1 and GLUT3, and sodium-dependent glucose symporters, such as SGLT1 and SGLT2, are overexpressed in cancers. As mentioned earlier, there are also glucose transporters that are downregulated in cancer cells, such as GLUT12 in neuroendocrine prostate cancer [47]. Decreased glucose uptake by cancer cells due to the inhibition of glucose transporters may be a potential therapeutic method in cancer diseases. An effect of a hypoxic condition is increased expression of GLUT1 and its increased transport of vesicles containing GLUT1 from the intracellular compartment into the cell membrane [48]. In this way, the uptake of glucose by cancer cells is increased, and cancer may develop and increase the risk of its metastasis. As mentioned earlier, this glucose transporter is involved in enhancing malignant potential and invasiveness in several cancers [49,50].

There are several methods for inhibition of glucose transporters, such as the use of glucose transporters’ inhibitors, small molecules, antibodies against glucose transporters, and siRNA. Inhibitors of GLUT1 were used, compounds such as STF-31 [28], phloretin [51], hydroxylated phenyl esters [52], and others [53] (Table 3). STF-31 belongs to the second class in the group of compounds pyridyl aniline thiazoles [54]. It is a small molecule, which was first used as selective target VHL-deficient renal cell carcinoma cells. This inhibitor inhibits cell proliferation and induces apoptosis in breast cancer cell lines. Obtained results revealed that this inhibitor blocks glucose uptake and/or glycolysis. It acts only on cancer cells and does not influence normal cells [28]. Researchers also observed that renal cell carcinoma is rescued from the cytotoxic effects of this inhibitor if these cells express a high level of GLUT2. This inhibitor selectively kills renal cell carcinomas cells by its binding to the GLUT1. Animal studies revealed that three daily intraperitoneal doses of a soluble analogue of STF-31 effectively reduce the growth of tumors of VHL-deficient renal cell carcinoma cells in these mice. Authors suggest that STF-31 is not restricted to VHL-restricted tumors, but it is lethal to cancers that have a high level of GLUT1 and require glycolysis [28]. The antiproliferative effect of metformin in breast cancer cell lines MDA-MB-23 may be potentiated by STF-31 [55]. It is suggested that the target spectrum of this inhibitor appears to be relatively narrow [56].

WZB117 is a small molecule which inhibits GLUT1 and cancer growth [52]. It acts on cancer cells in vitro and in vivo. Inhibition of GLUT1 by WZB117 decreases the levels of intracellular ATP and glycolytic enzymes. Animal studies performed on nude mice revealed that WZB117 inhibits cancer growth. After daily intraperitoneal injection of this inhibitor, the sizes of the compound-treated tumors were on average more than 70% smaller in comparison with control animals treated with PBS/DMSO. The inhibitory activity of WZB117 on cancer growth was also observed in experiments with human cancer cell lines, such as non-small cell lung cancer (NSCLC) (H1299 and A549) and human breast ductal carcinoma (MCF7). WZB117 used inhibits glucose transport in a dose-dependent manner and inhibits cancer cell growth. Researchers have observed similar results in experiments performed on human nontumorigenic NL20 lung and MCF12A breast cells. Authors also observed the synergistic anticancer effects between WZB117 and anticancer drugs cisplatin and paclitaxel [52]. A synergistic effect of WZB117 and gefitinib, the first selective inhibitor of the epidermal growth factor receptor’s (EGFR) tyrosine kinase domain, was demonstrated in animal studies [57]. It was found that GLUT1 is involved in gefitinib resistance of non-small cell lung cancer [57] and in radioresistance [58]. Inhibition of GLUT1 sensitizes NSCLC cells to gefitinib. The administration of WZB117 and gefitinib significantly inhibits the growth of tumors in animal models, in comparison with administration of both compounds alone. These results were confirmed in experiments with human NSCLC cell lines A549, H1299, PC-9, and HCC827. Authors also observed that inhibition of GLUT1 sensitizes pancreatic and ovarian cancer cells to gefitinib [57]. Inhibition of GLUT1 by WZB117 used also sensitizes colon and breast cancer cells to conventional chemotherapeutic agents and radiation [59,60,61]. Two other inhibitors of GLUT1 were also synthesized: WZB27 and WZB116. In the breast cancer cell line (MCF-7), these inhibitors reduce basal glucose uptake and cell proliferation, induce apoptosis, and arrest the cell cycle in the G1/S phase in lung, breast, colon, and cervical cancer cells. These compounds influence only cancer cells, without affecting a normal cell line MCF12A. These compounds demonstrate a synergistic effect with anticancer drugs cisplatin and paclitaxel [62].

A highly selective GLUT1 inhibitor is BAY-876. It is under preclinical study for oncolytic treatment [63]. This inhibitor decreases the uptake of glucose by triple-negative breast cancer cell lines [64]. Triple-negative breast cancer is a deadly form of breast cancer. It is highly malignant, invasive, and chemoresistant. This cancer is defined by the lack of estrogen receptor, progesterone receptor, and human epidermal growth factor receptor 2 expression [65].

Glutor is the next inhibitor of glucose transporters. One of the very potent glucose uptake inhibitor—piperazin—has been newly discovered [66]. Glutor targets GLUT1, GLUT2, and GLUT3 with low nanomolar potency. For example, IC_50_ is 4 nM and 6 nM for pancreatic and brain cancer cells. On the other hand, there are also cancer cells that were suppressed with IC < 100 nM. In several cancers, such as thyroid carcinomas, endometrial and breast cancer, pancreatic cancer, head and neck tumors, and non-small cell lung cancer, there are upregulated GLUT1 and GLUT3. Therefore, dual-specific GLUT1 and GLUT3 inhibitors may be required for cancer targeting. It is noteworthy that there are glucose uptake inhibitors targeting GLUT1 and GLUT3. An example may be chromophynones [67]. Researchers investigated the influence of Glutor on 94, mostly malignant, human cancer cell lines. Authors have found that Glutor inhibits glucose uptake, glycolysis, and efficiency suppresses the growth of various human cancer cell lines. Researchers also observed that inhibition of glucose uptake, which causes hypoglycemia conditions, induces overexpression of GLUT1, and especially GLUT3. Rapidly dividing cells, such as cancer cells, may use glutamine, as a source of C- and N-source. Decreased expression of GLUT1 sensitizes lung cancer cells to inhibition of glutamine utilization, causing apoptosis and decreased growth of cancer. Dual inhibition of glutaminolysis and glycolysis has positive therapeutic effects in the treatment of ovarian cancer [68]. Therefore, the administration of dual-specific inhibitors of GLUT1 and GLUT3 and an inhibitor of glutamine metabolism may synergistically inhibit cancer cell growth. Specific inhibition of GLUT1 and GLUT3 may be important, for example in human colorectal cancer. These glucose transporters are overexpressed in cells of this cancer, but GLUT3 plays a role distinct from that of GLUT1 in colorectal cancer. Under low-glucose conditions, GLUT3 is more important for colorectal cancer cells growth than GLUT1 [69].

There are also several other synthetic and natural inhibitors of glucose transporters. For example, fasentin, an inhibitor of GLUT1, which binds directly to GLUT1 and inhibits glucose uptake, increases apoptosis in prostate cancer, multiple myeloma cells, and acute promyelocytic leukemia cells. It sensitizes these cancer cells to FAS ligand-death receptor signaling [70].

Oxime-based GLUT1 inhibitors are a new group of inhibitors [71]. Their chemical structure differs from phloretin, WZB-117, and other inhibitors of GLUT1. Oxime-based inhibitors bind to GLUT1, inhibiting glucose transport and cell proliferation in H1299 lung cancer cells [71]. The basic structure of this group of inhibitors provides the basis for distinguishing the next generation of GLUT1 inhibitors [56]. Potent GLUT4-selective inhibitors that exhibit cytotoxicity in myeloma are also identified [72].

Polyphenols constitute a large family of natural compounds widely distributed in plants. These compounds protect against stressors, such as UV light and pests. According to their chemical structures, they may be divided into classes and subclasses [73]. They may protect directly against cancer through epigenetic modification and influence glucose uptake and metabolism in cancer cells. Polyphenols interfere with GLUT and SGLT proteins. Apigenin is a natural phytoestrogen flavonoid present in carious fruits, vegetables, bean, and tea. It significantly inhibits the expression of GLUT1 and enhances the chemosensitivity of laryngeal carcinoma HEp-2 cells to cisplatin [74]. Apigenin decreases the expression of GLUT1 at both the mRNA and protein, causing the inhibition of the proliferation of pancreatic cancer cells [75]. Several studies indicate that polyphenols decrease the uptake of glucose by breast cancer cell lines, such as MCF-7, MDA-MB-231, T47D, and others. For example, narigenin, a grapefruit flavone, inhibits basal and insulin-stimulated glucose uptake, causing inhibition of proliferation of cancer cells, resveratrol decreases the expression of GLUT1 in cancer, hespertin downregulates GLUT1 and impairs translocation of GLUT4 from an intracellular compartment into a plasma membrane, kaempferol decreases the expression of GLUT1, and so on [73]. There are also inhibitors specific for other glucose transporters. Phloretin is a natural compound existing in fruits, such as apples and pears. It is a GLUT2 inhibitor [76,77], which retards tumor growth and induces apoptosis in leukemia, melanoma, and colon cancer cells [78]. These effects are due to the inhibition of glucose transport by GLUT2 [76]. Phloretin also sensitizes cancer cells to paclitaxel [79]. Quercetin, a flavonoid compound of fruits, vegetables, and grains, is a GLUT2 inhibitor, causing inhibition of glucose absorption by this glucose transporter. It reduces the risk of lung cancer and other cancers [80]. The next natural flavonoid, silibinin, also known as silibin, is a GLUT4 inhibitor. Investigations of silibinin influence prostate cancer, which revealed its relative safety as an anticancer agent [81].

Ritonavir is the HIV protease inhibitor, which inhibits GLUT4. In melanoma cells, ritonavir inhibits glucose uptake by GLUT4 [82].

There are also inhibitors that do not interact with GLUT protein directly. For example, some of the DNA-damaging anticancer agents, such as adriamycin, camptothecin, and etoposide, decrease GLUT3 expression in HeLa cells, causing their deaths [83]. It was found that certain inhibitors of glycogen synthase kinase-3 (GSK-3) are also inhibitors of GLUT3. These inhibitors decrease the expression of GLUT3 at the transcription level; however, they do not inhibit GLUT3 directly. In this way, these inhibitors cause apoptotic cancer cell death [84].

MicroRNAs (MiRNAs) are short, non-coding RNAs that bind the 3′untranslated regions (3′-UTRs) of mRNAs. They act as negative post-transcriptionally regulators of gene expression. There are oncogenic miRNAs (onco-RNAs) that are upregulated in human cancers, stimulate cell proliferation, and inhibit apoptosis. There are also tumor-suppressive miRNAs that are down-regulated in cancer and prevent the development of cancer. There are results that use microRNA-195-5p, a microRNA targeting expression of GLUT3, inhibiting uptake of glucose and the growth of bladder cancer cells [85,86]. MicroRNA-7-5p suppresses oncogenes in the MCF-10A mammary epithelial cells [87]. MicroRNA, such as miR-125a-5p targeting the expression of GLUT1, plays a role as a tumor suppressor and regulator of glucose metabolism in several cancers, mainly in thyroid carcinoma [38]. For more details on microRNAs as anticancer molecules, see [88].

Another anticancer strategy is the administration of short hairpin RNA (shRNA). Silencing of GLUT1 expression with an shRNA decreases uptake of glucose in a triple-negative (MDA-MB-468 and Hs578T) and in SK-BR3 cell lines. This procedure also decreases the growth of MDA-MB-468 cells [89,90], and mouse mammary cell line 78617GL, both in vitro and in vivo [53]. Results obtained in other studies revealed that silencing of GLUT4 by a GLUT4 shRNA decreases glucose uptake in MCF-7 and MDA-MB-231 breast cancer cells, impairing cell proliferation and viability of these cancer cells [91].

Small interfering RNA (siRNA) is a class of double-stranded RNA non-coding RNA molecules. It is also known as short interfering RNA or silencing RNA. These molecules were used in experiments with non-small cell lung cancer cells [57]. Knockdown of GLUT1 decreases the levels of GLUT1 in gefitinib-resistant NSCL cells. Gefitinib is the standard of treatment for non-small cell lung cancer. Results obtained revealed that GLUT1 is involved in the gefitinib resistance of NSCL cancer cells. Researchers also observed that the genetic inhibition of GLUT1 sensitizes resistant NSCL cancer to gefitinib. Knockdown of GLUT1 showed a modest inhibition of cell growth. Further treatment of these cells by gefitinib decreased the number of viable cancer cells and increased the number of dead cells. Based on results obtained, researchers suggest that targeting GLUT1 may be a therapeutic approach to sensitizing resistant non-small cell lung cancer cells to gefitinib [57]. The role of GLUT1 in radioresistance of cancer cells was described by other authors [58]. It was also found that other glucose transporters, such as GLUT12, may be a potential therapeutic target in cancer treatment [92]. According to this result, another study investigated the administration of siRNA for GLUT12-knockdown in MCF-7 human breast cancer cells [93]. Results obtained revealed that GLUT12 knockdown significantly reduces the level of GLUT12 in these cells. This observation is very important because GLUT12 plays a role in high glucose-induced cell migration, and, therefore, according to the authors’ suggestion, it may be a new strategy for the diagnosis and therapy of breast cancer in patients with hyperglycemia, as is observed, for example, in diabetes mellitus [93].

Another anticancer strategy is the transfection of antisense cDNA into cancer cells. This procedure causes the expression of cancer cell growth in vitro and tumor growth in vivo, and reduces the invasiveness of cells [94]. Results obtained in another study showed that transfection of GLUT1 antisense cDNA reduces the level of GLUT1 mRNA and cell proliferation in human leukemia cells [95], decreases glucose uptake and GLUT1 mRNA levels in MKN45 gastric cancer cell line [96], reduces in vitro invasiveness of rhabdosarcoma and glioblastoma cell line [94], and decreases glucose uptake and the level of GLUT1 mRNA and GLUT1 protein, as well as decreasing proliferation of HEp-2 laryngeal carcinoma cells [49]. Another study revealed that GLUT5 knockdown by antisense oligonucleotide decreases the uptake of glucose and proliferation of MCF-7 and MDA-MB-231 breast cancer cells [97].

Decreased uptake of glucose into cancer cells may also be due after treatment of these cells by antibodies anti-GLUT1. This procedure inhibits the growth of head and neck squamous cell cancer cells (Cal27), induces their apoptosis, and sensitizes these cells to chemotherapy (cisplatin) [98]. Other experiments revealed that antibodies anti-GLUT1 inhibit proliferation by 50% in the non-small cell lung carcinoma, and by 75% in breast cancer cell lines. In these cells, induction of apoptosis was observed [99]. Administration of anti-GLUT1 potentiates the antiproliferative effects of cisplatin, paclitaxel, and gefitinib.

The other anticancer therapy is administration of carbohydrate-drug conjugates. Glycoconjugates are designed for selective uptake by cancer cells that overexpress glucose transporters. The first glycoconjugate for GLUT1 was glufosfamide in 1995 [100]. D-19575 is a glucose derivative of ifosfamide mustard. In animal models, it shows a broad spectrum of antitumor activity. Researchers observed an increase in the cancer-selective uptake of D-19575 by GLUT1. After hydrolysis or glucosidase-mediated cleavage of glucose in cancer cells, the phosphoramide mustard is liberated, which is the anticancer active drug [100]. Following the introduction of glufosfamide, several glycoconjugates for GLUT targeting were synthesized. An example of these cytotoxic molecules may be chlorambucil, methane sulfonate, paclitaxel, and others. Other glucose transporters are investigated, such as GLUT2, GLUT3, and GLUT12, as carrier proteins for conjugates [101]. Adriamycin (doxorubicin) conjugated with a glucose analogue and succinic acid is designed to target cancer cells through GLUT1. Its influence on several cancer cells was investigated, such as MCF-7 and MDA-MB-231 cell lines. Results obtained revealed better inhibition of cancer cells and lower toxicity to normal cells [102]. However, it is effective against different solid tumors in clinical applications, and its use is limited due to systemic toxicity and multidrug resistance [53]. Paclitaxel, another glycoconjugant, is widely used for the treatment of breast, ovarian, and lung carcinomas. Its clinical application is reduced due to its low water solubility [103]. Oxiplatin is a platinum antitumor prodrug, which is commonly used as a chemotherapeutic agent. It was investigated in experiments with cancer cells, such as human colon cancer (HT29) and breast cancer (MCF-7). Its multiple side effects limit its use [104].

### 3.3. Sodium-Dependent Glucose Cotransporters as a Target in Anticancer Therapy

An increased level of SGLTs is also used by cancer cells to enhance their glucose uptake. Sodium-dependent glucose cotransporters may also be a target therapy in anticancer therapeutic strategies [105]. Like in the case of GLUT proteins, inhibitors of SGLTs may be used as a strategy in cancer treatment [106,107,108]. There are several new antidiabetic drugs that inhibit SGLTs, such as phlorizin, empagliflozin, dapagliflozin, and canagliflozin [108]. Animal studies showed that administration of dapagliflozin decreases the uptake of a specific SGLT tracer, Me-4FDG in mice with pancreatic cancer [107]. Researchers also reported that canagliflozin and dapagliflozin decrease cancer viability in the mouse xenograft model, and canagliflozin reduces tumor growth by increasing necrosis. Based on results obtained, researchers suggest that SGLT2 inhibitors may be anticancer drugs used in pancreatic cancer [107]. It is also suggested that inhibitors of SGLT1 and SGLT2 may be used in anticancer therapy in the case of prostate, ovarian, and brain cancers [108]. Inhibition of SGLT2 may be therapeutic targets for early-stage lung adenocarcinoma (LADC) [109]. These studies were performed on samples performed from patients with atypical adenomatous hyperplasia (AAH), which is a precursor of LADC, and on mouse models, in which lung tumors were induced by intranasal administration of Adeno-Cre. Results obtained showed that selective targeting of SGLT2 with inhibitor canagliflozin greatly decreases tumor growth and increases survival in animal models and patients-derived xenografts of LADC. The authors suggest that targeting SGLT2 in lung cancer may decrease lung cancer progression at early stages of development [109].

Thyroid cancer therapy is based on surgery followed by radioiodine treatment. NIS, a sodium iodide symporter, is involved in the incorporation of radioiodine by cancer cells. Expression of NIS is cancer cells allows for the therapeutic application of radioactive substrates of NIS, such as ^123^I, ^124^I, and ^127^I. It is suggested that NIS expressed in thyroid primary tumors may be helpful in anticancer therapy [110,111]. There are suggestions that decreased NIS levels account for the reduced activity of iodide in thyroid cancers [112,113]. Therefore, to achieve radioactivity uptake into cancer cells sufficient for therapy, stimulation of NIS expression is necessary. This is due to high levels of thyroid-stimulating hormone (TSH). This procedure is very important because selective induction of iodide uptake is necessary to target of cancers with radioiodine [114,115]. After total thyroidectomy, to target remnant thyroid cancer and metastasis, β-emiting radioactive iodide is used. NIS is also expressed in the majority of breast cancer. Unfortunately, its low level in these cancer cells is insufficient for radioiodine therapy. Therefore, as a potent inducer, retinoic acid is used [114]. Results obtained results suggest that radioiodine may be used as an adjuvant treatment in breast cancer [116].

## 4. Conclusions

Glucose metabolism in cancer cells differs from this process in normal cells. Cancer cells prefer the process of glycolysis, which generates less molecules of ATP than complete oxidative breakdown; therefore, cancer cells need more molecules of glucose. Upregulation of glucose transporters in cancer cells increases uptake of glucose; therefore, several glucose transporters are overexpressed in cancer cells. GLUT1 is especially overexpressed in cancer cells; however, upregulation is also observed in the case of other glucose transporters, such as GLUT3 and NIS. Changes of immunostaining intensity, observed in investigated glucose transporters, may characterize the development, stage, and type of cancer. Therefore, in many cancers, the overexpression of GLUT1 or GLUT3 may be treated as a marker of the stage of carcinogenesis, aggressiveness of cancer, as well as prognosis and OS for patients. There are also results revealed that glucose transporters may be involved in development, progression, and metastasis of cancer. Therefore, researchers postulate, and there are obtained results that suggest, that administration of glucose transporters’ inhibitors may be used in anticancer therapy. There are also cases described in which upregulation of glucose transporter, as for example NIS, which is used in radioiodine therapy, can also help patients with cancers.

## Figures and Tables

**Table 1 cancers-13-04184-t001:** Characteristics of the human sodium-independent glucose transporter (GLUT) proteins [1].

Gene	Transporter	Localization and Characteristics
*SLC2A1* 1p35-p31.3.	GLUT1 It is composed of 492 amino acids with a molecular weight of approximately 54.2 kDa. On SDS PAGE, GLUT1 runs as a band between 50–60 kDa, due to the N-linked glycosylation at ASN-45.	The larger molecular weight GLUT1 is present in the cerebral cortex and microvessels, the lower isoform is present in the microvessels-depleted brain membranes (neuronal/glial membranes) and synaptosomes, and an intermediate form is present in the choroid plexus. It is also present in erythrocytes, granulocytes, adipocytes, muscle cells, kidney, colon, and many fetal tissues. It acts as a receptor for human T cell leukemia virus type 1. It plays an essential role in CD4+ T cell activation. It has affinity for glucose, and also transports galactose, mannose, and glucosamine.
*SLC2A2* 3q26.1-q26.2.	GLUT2 It is composed of 524 amino acids with a molecular weight of 58 kDa.	It is expressed in the intestinal absorptive epithelial cells, hepatocytes, pancreatic β-cells, and kidney cells. GLUT2 mRNAs are present in the brain nuclei, including the nucleus tractus solitatius, the motor nucleus of the vagus, the paraventricular hypothalamic nucleus, the lateral hypothalamic area, the arcuate nucleus, and the olfactory bulbs. GLUT2 is a low affinity transporter for glucose, galactose, mannose, and fructose. It is a high affinity transporter for glucosamine.
*SLC2A3* 12p13.31 It is copy number variable in humans. A deletion or duplication of 129 kb results in variants of one or three (or more) total *SLC2A3* gene copies.	GLUT3 It is composed of 496 amino acids with predicted weight 54 kDa.	It is highly expressed in the brain, spermatozoa, placenta, preimplantation embryos, fibroblasts human platelets, retinal endothelial cells, and white blood cells.
*SLC2A4* 17p13	GLUT4 It is composed of 509 amino acids with a molecular weight of approximately 55 kDa.	It is expressed in the skeletal muscle cells, cardiomyocytes, adipose tissue, hypothalamus, cerebellum, cortex, and hippocampus. GLUT4 is a high affinity transporter of glucose. It also transports dehydroascorbic acid and glucosamine.
*SLC2A5* 1p36.2.	GLUT5 It is composed of 501 amino acids with a molecular weight of 43 kDa.	It is expressed in the apical membrane of the enterocytes, in the plasma membrane of mature spermatozoa, kidney, fat, skeletal muscle, and the brain.
*SLC2A6* 9q34	GLUT6 (formerly designated GLUT9). It is composed of 507 amino acids with a predicted weight of 46 kDa.	It is predominantly expressed in the brain, spleen, peripheral leukocytes, and in germinal cells of the testis. It exhibits glucose transport activity; however, it appears to be a low-affinity facilitator of glucose.
*SLC2A7* 1p36.22.	GLUT7 It is composed of 524 amino acids.	It is primarily expressed in the small intestine (distal region), colon, testis, and the prostate gland. It has a high affinity for both glucose and fructose. It does not transport galactose, 2-deoxy-D-glucose and xylose.
*SLC2A8* 9q33.3	GLUT8 It is composed of 477 amino acids. It migrates as a 35-kDa protein.	It is expressed in the testis, cerebellum, adrenal gland, liver, spleen, brown adipose tissue, in the spermatocytes and in the human mature spermatozoa. It is a high affinity transporter of glucose.
*SLC2A9* 4p15.3–p16 This gene consists of 12 exons for GLUT9a or of 13 exons for GLUT9b isoform.	GLUT9 (earlier designated as GLUTX). It is composed of 540 amino acids (the major isoform of GLUT9a) or of 512 amino acids (the GLUT9b isoform).	GLUT9b is expressed only in the liver and kidney, whereas GLUT9a is present in many more tissues and organs, such as liver, kidney, intestine, leukocytes, and chondrocytes. It is a glucose, fructose, and urate transporter. GLUT9a and GLUT9b transport urate with the same kinetics.
*SLC2A10* 20q13.1	GLUT10 It is composed of 541 amino acids.	It is expressed in the skeletal muscle, heart, and adipose tissue. GLUT10 mRNA is expressed in the liver, pancreas, placenta, and kidney. GLUT10 is has a very high affinity for both deoxy-D-glucose and D-galactose, but not for fructose.
*SLC2A11* 22q11.2 Gene contains of 10–13 exons.	GLUT11 (formerly designated GLUT10) Three isoforms of GLUT11 have been cloned: GLUT11-A, GLUT11-B, and GLUT11-C. It is composed of 496 amino acids.	GLUT11-A is expressed in the heart, skeletal muscle, and kidneys; GLUT11-B is expressed in the kidneys, adipose tissue, and placenta; GLUT11-C is expressed in the adipose tissue, heart, skeletal muscle, and pancreas. GLUT11 appears to recognize fructose with comparable affinity to, possibly with even higher affinity than, glucose (Km ~0.2 mM) but not galactose.
*SLC2A12* 6q23.2.	GLUT12 (formerly designated GLUT8) It is composed of 617 amino acids.	It is expressed in the heart, skeletal muscle, adipose tissue, prostate gland, kidneys, small intestine, chondrocytes, and in the placenta. It exhibits transport activity for glucose, galactose, and fructose.
*SLC2A13* 22q12	GLUT13 (HMIT) Human HMIT is composed of 629 amino acids with a molecular weight of 83 kDa.	It is expressed in the brain, white and brown adipose tissue, and the kidneys. It is a H^+^/*myo*-inositol cotransporter that exhibits transport activity only for *myo*-inositol with a high affinity.
*SLC2A14* 12p13.3. It exhibits a genomic organization similar to that of GLUT3	GLUT14 It has two alternatively spliced forms. GLUT14-S contains 10 exons and produces a 497 amino acid protein. GLUT14-L has an additional exon and codes for protein with 520 amino acids.	Both isoforms of GLUT14 are specifically expressed in the human testis.

**Table 2 cancers-13-04184-t002:** Characteristics of the human sodium-dependent symporters [1].

Gene	Protein	Localization and Characteristics
*SLC5A1* 22q12.3	SGLT1 It is composed of 664 (662) amino acids with a molecular weight of 73 kDa.	It is expressed in the mature enterocytes of brush border membrane in the small intestine, trachea, prostate, heart, and kidneys and in the luminal membrane of intracerebral capillary endothelial cells. SGLT1 mRNA is detected in the human prostate, testis, trachea, and uterus (cervix). SGLT1 is a high-affinity, low-capacity transporter for D-glucose and D-galactose. It does not transport fructose, mannose, and xylose. It may also behave as the glucose receptor in the heart and brain.
*SLC5A2* 16p11.2	SGLT2 It is composed of 672 amino acids with a predicted mass of 73 kDa.	It is expressed on the apical membrane of renal convoluted proximal tubules. SGLT2 mRNA was detected in the mammary glands, liver, lungs, intestine, skeletal muscle, and spleen. It is suggested that SGLT2 may also behave as the glucose receptor in the heart and brain. SGLT2 represents a low-affinity, high-capacity sodium-glucose symporter for glucose.
*SLC5A3* 21q22.11 Splicing within and distal to exon 2, leads to 3 transcripts namely SMIT1, SMIT2 and SMIT3.	SMIT1	It is expressed in the kidneys, brain, placenta, pancreas, heart, skeletal muscle, and the lungs. Its mRNA was detected in the blood vessels (choroid plexus and coronary artery, intestine, ovary, pineal gland, prostate, thyroid gland, and uterus (cervix). It transports *myo*-inositol, L-fucose, and L-xylose (but not their D-isomers) and does not distinguish between D- and L-glucose.
*SLC5A4* 22q12.3.	SGLT3 (SAAT1) It is composed of 659 amino acids.	It is expressed in the kidneys, uterus, testis, intestinal autonomic nervous, skeletal muscle, brain, proximal tubule of human kidneys and in the cholinergic neurons in the enteric nervous system, and its mRNA is detected in the pancreas, lungs, and liver. SGLT3 demonstrates a lack of sugar transport activity. It does not transport glucose, but it is a glucose sensor.
*SLC5A5* 19p13.11.	NIS	It is a Na^+^/iodide cotransporter. It is principally expressed in the thyroid, where it is responsible for the accumulation of iodide necessary for thyroid hormones T_3_ and T_4_. NIS is also expressed in the lactating breast, colon, stomach, and ovary. The substrate specificities for NIS are I^−^ (ClO_4_^−^, SCN^−^, NO_3_^−^, Br^−^).
*SLC5A6* 2q12.	SMVT	It is a multivitamin Na^+^ cotransporter. It is expressed in the brain, heart, kidneys, lungs, and placenta. It mediates Na^+^-dependent uptake of vitamins such as pantothenic acid, biotin, and α-lipoic acid. Because SMVT has a high sequence identity and similarity with NIS, it also behaves as a Na^+^/iodide cotransporter
*SLC5A7* 2q13.	CHT1	It is a Cl^−^–dependent Na^+^/choline cotransporter that is expressed in the central nervous system in the spinal cord and medulla and transport is sensitive to pH.
*SLC5A8* 12q23.1.	SMCT1	It is expressed in the small intestine, kidneys, brain, retina, and muscle. It is a monocarboxylate cotransporter which transports lactate, pyruvate, and nicotinate with a stoichiometry of 2:1.
*SLC5A9* 1p32.	SGLT4	It is expressed in the small intestine, kidneys, brain, liver, heart, uterus, and lungs. SGLT4 mRNA, at low levels, was also detected in the testis, pancreas, and skeletal muscle. It is involved in absorption and/or the reabsorption of D-mannose, D-fructose, and D-glucose.
*SLC5A10* 17p11.2 There are four splice variants.	SGLT5	It is expressed in the human kidney cortex. SGLT5 mRNA was detected also in the left atrium of the heart, ovary, skin (foreskin), testis, and in vas deferens. SGLT5 has been reported to be a sodium-dependent sugar transporter with a relatively high affinity and capacity for mannose and fructose relative to glucose and galactose.
*SLC5A11* 16p12.1.	SGLT6 (now known as SMIT2).	It is detected in the brain, heart, kidneys, skeletal muscle, spleen, liver, placenta, lungs, leukocytes, and neurons. SGLT6 transports *myo*-inositol and d-chiro-inositol. It shows stereoscopic transport of D-glucose and D-xylose without affinity for fructose. In humans, *SLC5A11* interacts with immune-related gene(s) and may function as an autoimmune modifier gene.
*SLC5A12* 11p14.2.	SMCT2.	It is expressed in the small intestine, kidneys, brain, retina, and muscle. It is a monocarboxylate cotransporter that transports lactate, pyruvate, and nicotinate with a stoichiometry of 2:1.

**Table 3 cancers-13-04184-t003:** Glucose transporters as target in anticancer therapy.

Methods for Inhibition of Glucose Transporters	Therapeutic Effects and Mode of Action
Inhibitors of GLUTs	
(1) WZB117	(1) It inhibits cancer growth and sensitizes colon and breast cancer cells to conventional chemotherapeutics agents and radiation. It also inhibits GLUT1 and glucose transport into cancer cells, decreases the levels of intracellular ATP and glycolytic enzymes.
(2) WZB27 and WZB116	(2) These inhibitors reduce basal glucose uptake and cell proliferation, induce apoptosis, and arrest the cell cycle in G1/S phase.
(3) BAY-876	(3) This inhibitor decreases uptake of glucose by a triple-negative breast cancer cell lines.
(4) Glutor	(4) It targets GLUT1, GLUT2, and GLUT3. Glutator inhibits glucose uptake, glycolysis, and efficiency suppresses the growth of various human cancer cell lines. Dual inhibition of glutaminolysis and glycolysis has a positive therapeutic effect in the treatment of ovarian cancer, and the administration of a dual-specific inhibitor of GLUT1 and GLUT3 and the inhibitor of glutamine metabolism may synergistically inhibit cancer cell growth.
(5) Fasentin	(5) Inhibitor of GLUT1, which binds directly to GLUT1 and inhibits glucose uptake, increases apoptosis in prostate cancer, multiple myeloma cells, and acute promyelocytic leukemia cells. It sensitizes theses cancer cells to FAS ligand-death receptor signaling.
(6) Oxime-based inhibitors	(6) They bind to GLUT1, inhibiting glucose transport and cell proliferation in H1299 lung cancer cells.
(7) Polyphenols	(7) Apigenin significantly inhibits the expression of GLUT1 and enhances the chemosensitivity of laryngeal carcinoma HEp-2 cells to cisplatin, inhibiting proliferation of pancreatic cancer cells. Narigenin inhibits basal and insulin-stimulated glucose uptake, causing inhibition of proliferation of cancer cells. Resveratrol decreases expression of GLUT1 in cancer. Hespertin downregulates GLUT1 and impairs translocation of GLUT4 from intracellular compartment into the plasma membrane. Kaempferol decreases expression of GLUT1. Phloretin is a GLUT2 inhibitor, which retards tumor growth and induces apoptosis in leukemia, melanoma, and colon cancer cells, and also sensitizes cancer cells to paclitaxel. Quercetin is a GLUT2 inhibitor, causing inhibition of glucose absorption by this glucose transporter. It reduces the risk of lung cancer and other cancers. Silibinin, also known as silibin, is a GLUT4 inhibitor. Investigations of silibinin influence on prostate cancer revealed its relative safety as an anticancer agent.
(8) Adriamycin, camptothecin, and etoposide	(8) decrease of GLUT3 expression in HeLa cells, causing their deaths. They do not interact with GLUT protein directly; they are DNA-damaging anticancer agents.
(9) STF-31	(9) It was first used as selective target VHL-deficient renal cell carcinoma cells. This inhibitor inhibits cell proliferation and induces apoptosis in breast cancer cell lines, blocks glucose uptake, and/or glycolysis. It acts only on cancer cells, and does not influence normal cells. Renal cell carcinoma is rescued from the cytotoxic effects of this inhibitor if these cells express a high level of GLUT2. This inhibitor selectively kills renal cell carcinomas cells by its binding to the GLUT1.
MicroRNAs (MiRNAs)	
They act as negative post-transcriptionally regulators of gene expression	MicroRNA-195-5p targeting expression of GLUT3, inhibits uptake of glucose and growth of bladder cancer cells. MicroRNA-7-5p suppresses oncogenes in the MCF-10A mammary epithelial cells. MicroRNA-125a-5p targeting expression of GLUT1, plays a role as tumor suppressor and regulator of glucose metabolism in several cancers, mainly in thyroid carcinoma.
Short hairpin RNA (shRNA)	
Their role is the silencing of glucose transporters	Silencing of GLUT1 expression with an shRNA decreases uptake of glucose in a triple-negative (MDA-MB-468 and Hs578T) and in SK-BR3 cell lines, and also decreases the growth of MDA-MB-468 cells. Silencing of GLUT4 by a GLUT4 shRNA decreases glucose uptake in MCF-7 and MDA-MB-231 breast cancer cells, impairing cell proliferation and viability of these cancer cells.
Antisense cDNA	
Transfection of antisense cDNA into cancer cells	This procedure reduces invasiveness of cancer cells, transfection of GLUT1 antisense cDNA reduces the level of GLUT1 mRNA and cell proliferation in human leukemia cells, decreases glucose uptake and GLUT1 mRNA level in the MKN45 gastric cancer cell line, reduces invasiveness of rhabdosarcoma and glioblastoma cell lines, decreases glucose uptake and level of GLUT1 mRNA and GLUT1 protein, as well as decreases proliferation of HEp-2 laryngeal carcinoma cells. GLUT5 knockdown by antisense oligonucleotide decreases the uptake of glucose and proliferation of MCF-7 and MDA-MB-231 breast cancer cells.
Antibodies anti-GLUT1	
	This procedure inhibits the growth of head and neck squamous cell cancer cells (Cal27), induces their apoptosis, and sensitizes these cells to chemotherapy (cisplatin), inhibiting proliferation by 50% in the non-small cell lung carcinoma, and by 75% in breast cancer cell lines. In these cells, the induction of apoptosis was observed. Administration of anti-GLUT1 potentiates the antiproliferative effects of cisplatin, paclitaxel, and gefitinib.
Carbohydrate-drug conjugates (Glycoconjugates)	
	Glufosfamide shows a broad spectrum of antitumor activity. After hydrolysis or glucosidase-mediated cleavage of glucose in cancer cells, the phosphoramide mustard is liberated, which is the anticancer active drug. Several glycoconjugates for GLUT targeting were synthesized, as an example of these cytotoxic molecules may be chlorambucil, methane sulfonate, paclitaxel, and others. In addition, other glucose transporters are investigated, such as GLUT2, GLUT3, and GLUT12, as carrier proteins for conjugates. Adriamycin (doxorubicin) conjugated with a glucose analogue and succinic acid is designed to target cancer cells through GLUT1. It inhibits cancer cells. However, it is effective against different solid tumors in clinical applications, and its use is limited due to systemic toxicity and multidrug resistance. Paclitaxel is widely used for the treatment of breast, ovarian, and lung carcinomas. Its clinical application is reduced due to its low water solubility. Oxiplatin is a platinum antitumor prodrug, which is commonly used as a chemotherapeutic agent. It was investigated in experiments with cancer cells, such as human colon cancer (HT29) and breast cancer (MCF-7). Its multiple side effects limit its use.
Inhibitors of SGLTs	
	There are several new antidiabetic drugs that inhibit SGLTs. Dapagliflozin decreases uptake of specific SGLT tracers, Me-4FDG in mice with pancreatic cancer. Canagliflozin and dapagliflozin decrease cancer viability in the mouse xenograft model, and canagliflozin reduces tumor growth by increasing necrosis. SGLT2 inhibitors may be anticancer drugs used in pancreatic cancer. Inhibitors of SGLT1 and SGLT2 may be used in anticancer therapy in the case of prostate, ovarian, and brain cancers. Inhibition of SGLT2 may be a therapeutic target for early-stage lung adenocarcinoma (LADC). Selective targeting of SGLT2 with inhibitor canagliflozin greatly decreases tumor growth and increases survival in animal models, and patients-derived xenografts of LADC. Targeting SGLT2 in lung cancer may decrease lung cancer progression in the early stages of development.

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
