# Peer review of "Glucose Transporters as a Target for Anticancer Therapy"

_cancers, 2021, doi:10.3390/cancers13164184_

Round 1

Reviewer 1 Report

The authors give an overview about glucose-transport-molecules in cancer, including SWEET - a new class of this transporters. The second part of this review deals with different types of glucose-transport-inhibitors.

The review is well written - only the part of Oral Squamous Cell Carcinoma (OSCC) can be improved: there are more papers dealing with the influence of GLUT-1 on the overall prognosis in OSCC. Moreover, glutaminolysis is directly linked to glycolyis - the articles of Spinelli JB et al. Science 2017, 358, 941-6 and Kappler M et al.  Int J Mol Sci 2019, 20, 4742 would enhance this part of the review.

Author Response

Dear Reviewer

Thank you very much for your review, opinion and suggestions. In regard to role of GLUT1 in OSCC we have problem. Earlier, I wrote an article on role of glucose transporters as markers in diagnosis and prognosis in cancer disease, and I wrote also on OSCC and GLUT1. In presented manuscript, we describe the role of glucose transporters as target in anticancer therapy. Therefore, we don't write on GLUT1 as marker in OSCC. But we included the role of glutamine in cancer metabolism, according to your suggestion. It was also suggestion by one Reviewer, that manuscript is too long and should be shortened. And again thank you very much your opinion and suggestions.

Reviewer 2 Report

This systematic review by Monika Pliszka is a comprehensive and state-of-the-art presentation of the new insight into the glucose transport and metabolism are essential for the posttreatment survival of tumor cells, leading to poor prognosis. Their approach entails a new and substantial contribution to current literature on the subject matter; however, I have the following major but serious concerns:

  1. This review study is largely confirmatory of a previously published study by 2007;53(4):233-56.: Int J Endocrinol. 2010;2010:205357.; Biochim Biophys Acta. 2012 Dec;1826(2):370-84., Preclinical studies underscore the dysregulation of glucose transport and energy metabolism pathways by oncogenes and lost functions of the tumor suppressors have been implicated as important biomarkers for cancer detection and as valuable targets for the development of new anticancer therapies, and therefore lacks significant novelty of this review.

  1. This review is too long and, in many places repetitive. Authors should review the paper content for redundancy and ensure only essentials are left.

  1. Though there is not standard limit for references in review articles, i feel over 115 references is just too many. Authors should look at cutting this down to < 100.

  1. There are also a few errors in English language grammar that require the authors' attention. 

  1. All abbreviations must be defined when they are first used.

  1. Authors may want to provide a more representative Graphical Abstract

  1. This manuscript cannot be accepted in its present form, I recommend REJECTION

Author Response

Dear Reviewer

Thank you very much for your review, opinion and suggestions.

Ad. 1. You are right that our manuscript is "largely confirmation of a previous study" But we think that there are novelity of this article. For example, in article by Airley and Mobasheri, which I have cited in my other articel, Authors describe especially role of HIF-1, GLUT1 and GLUT3. We describe also other glucose transporters and anticancer therapy. In the article by Calvo et al. (2010) is described, for example, the role of GLUT12. In our manuscript, we included short informations on all GLUT proteins, all SGLTs, and on new class, SWEET. We described more methods in anticancer therapy. In comparison with 3rd article, our article contains several other informations. In 2013 my article on expression of glucose transporters in cancer was published in BBA Rev. Canc. To date, it was cited 261 times (14 times in 2021) - according to Web of Science, it is highly cited article. However, it is cited in several articles, in these articles are also novelity, not only from my article.

Ad.2, 3. May be, article is too long, but we don't see informations that should be excluded. But, according to suggestion of other Reviewer, it was necessary to add few, additional informations. My last article on glucose transporters, published few days ago IJMS is longer as compared to reviewed manuscript and contains much more references (163). All cited articles, according to Reviewers, were necessary.

Ad.4 According to your suggestion, our manuscript was improved by the editing service.

Ad. 5. According to your suggestion, all abbreviations are defined when they are first used.

Ad.6. I'm sorry, we don't know, what to do.

Ad.7. Again thank you very much for your opinion and suggestions. We think, that, if article will be accepted for publication, it will be more friendly for readers.

Reviewer 3 Report

This review article is certainly dealing with an important and an interest-attracting issue for the readers of “Cancers” written by the authors who are specialists of this field.

Concerns:

  1. Overall review and some revisions by an oncologist or a pathologist would be appreciated, because there are some inaccurate descriptions or terminology in oncology especially in the first paragraph of the Introduction. In other parts, for example, page 6, line 243-244, the authors described as “Triple-negative breast cancer is a deadly form of breast cancer. It is a highly malignant, invasive, and chemoresistant.”, but it is generally known that the TNBC responds well to the initial neoadjuvant chemotherapy.

  1. There are many preceding excellent reviews for glucose transporters themselves including the authors’ own ones. Therefore, in this review, the authors should more concentrate on the description and discussion on the glucose transporters and therapeutic reagents for them in relation to cancer therapy with the negative aspects (predicted adverse effects, resistance due to compensation by transporters other than the targets, etc.).

  1. Checking typographical errors and mistakes in English grammar would be needed.
    For example, page 3, line 103: class II may be class III.

Author Response

Dear Reviewer

Thank you very much for your review, opinion and suggestions. With regards to your comments:

Ad. 1. We aren't oncologists and we aren't pathologists. Therefore, in several cases we asked specialists with request on help. In few cases we have received an answer. Unfortunately, there were also answers: "on this subject there are different informations (suggestions, results, etc). Because our manuscript is a review, it was wrote on the basis of literature. In all cases we cite an article. Therefore, according to your suggestion, may be inaccurate descriptions or oncological terminology, hower, they are according to cited literature. As in the case of "triple-negative breast cancer" [69]

Ad. 2. You are right, there are many articles, especially case reports or experimental, but much less of reviews. Aim of our manuscript was "potential role of glucose transporters as target of anticancer therapy". Therefore, we describe different suggested therapies, what was our main problem. Therefore, negative aspects of particular therapies were less important. Also, it was suggested by one Reviewer, that manuscript is too long and we should cut it.

Ad. 3. According to your suggestion, our manuscript was improved by the editing service.

Dear Reviewer, and again thank you very much for your opinion and suggestions. If manuscript will be accepted for publication, it will be more friendly for readers.

Round 2

Reviewer 3 Report

Concerning your Ad.1, I believe that the authors are responsible to orchestrate the content of their review paper which gathers information from multiple papers. Concerning Ad.2, I don't agree the authors' response that "Therefore, negative aspects of particular therapies were less important. ".  However, I also certainly understand the authors' points.